# Regulating Imbalanced Deep Models with User-Specified Metrics

## Abstract

Deep learning models implemented in real-world applications still face challenges from imbalanced data. Existing methods address the imbalance problem by balancing the models between the minority class and the majority class. However, practical applications may require an imbalanced optimization strategy that selectively unbalances the models and makes them more suitable for the applications than the balanced models. In this work, we first give a formal definition to accurately quantify the degree of imbalance of a model. Then, we propose a bias adjustment method that can efficiently optimize the model to a specified imbalance state according to application metrics or requirements so that this method has wide applicability. Finally, we introduce a training strategy that is advantageous to select the optimal representation parameters of the model during traditional training process. Extensive experiments verify the effectiveness and efficiency of our method, and compared with state-of-the-art algorithms, our method has significant improvement in different metrics including accuracy, F1 value and G-means.

## 1 Introduction

Deep learning has achieved enormous success in various fields, but it also faces a challenge due to imbalanced data. In fact, the datasets for many applications are imbalanced, where the majority class dominates most of the data while the minority class has few samples. The ratio of their sample sizes may be many orders of magnitude. A deep network model trained on such an imbalanced dataset will be seriously biased towards the majority class, resulting in misclassification of the minority class samples. This class-imbalance problem appears in many applications, such as sentiment classification (Wang et al., 2021), Twitter spam detection (Li & Liu, 2018), object detection (Oksuz et al., 2020) and medical science (Khushi et al., 2021).

There are many works to solve the imbalance problems in deep learning, which design different optimization objectives to balance the models to improve the performance of the minority class. In this paper, we call the above objectives as balanced optimization objectives. These works focus on a common imbalance scenario where the training data is unbalanced due to manual sampling errors or the scarcity of sampled objects but the actual test set is balanced, e.g. the long-tailed problem in image classification (Tan et al., 2020). This balance effect is shown on data(1) of Figure 1. The circles and pentagrams represent the samples of majority and minority classes, respectively. The lines represent the boundary of the models. The samples above and below the lines are predicted into the majority class and the minority class respectively. In Figure 1 (1), the green line represents the best boundary line. It means that the model trained on the unbalanced dataset can achieve the best accuracy on a balanced dataset. These research works mainly include re-sampling (Chawla et al., 2002; Liang et al., 2022), class-level or instance-level re-weighting (Lin et al., 2017; Liu et al., 2021), and two-stage methods (Wahab et al., 2017; Guo et al., 2022). Re-sampling uses down/up-sampling to obtain a balanced dataset and optimize a model on this dataset (Drummond et al., 2003; Barandela et al., 2004). Recently, re-weighting methods learn instance-level weight values with a balanced dataset, so that the models guided by these weights can achieve optimal performance on balanced test datasets (Ren et al., 2018; Hu et al., 2019; Liu et al., 2021; Guo et al., 2022). In two-stage methods, the models are also corrected by class-balanced optimization strategies at the second stage(Kang et al., 2019).

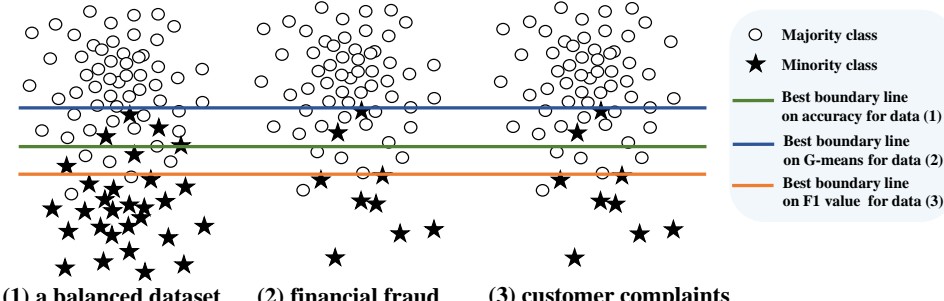

Figure 1: On data(1), the green line equally divides the circles and pentagrams and has the best accuracy. On data(2), G-means considers the recall of the minority and majority class, so the blue line is the best due to recognizing all pentagrams without significantly reducing the majority class recall. On data(3), F1 value takes into account the recall and precision of the minority class, so orange line is the best due to improving the recall of the pentagrams while not overly misclassifying the circles.

However, in many real-world applications, the online test data is also unbalanced like the training data and these applications have a certain preference among classes. We roughly divide them into 2 cases. In the first case, the minority class is more important than the majority class. For example, in financial fraud detection (Priscilla & Prabha, 2020; Warghade et al., 2020), fraudulent customers (i.e. minority class) are much more valued than normal customers (i.e. majority class), and the detection system is unwilling to omit any customer who may be fraudulent. Data(2) in Figure 1 shows this situation. The blue line as the best boundary identifies all fraudulent customers (pentagrams) and G-means is the metric for fraud detection (Sisodia et al., 2017). Similar situations appear in disease detection (Cui et al., 2020), information security (Shu et al., 2022), crime prediction (Hossain et al., 2020), etc. In the second case, the majority class is more critical. For example, in customer complaints recognition, the system pays more attention to major incidents (i.e. majority class) but minor incidents (i.e. minority class) cannot be ignored either. Thus, while improving the recall of the minor incidents, the system does not want to excessively misclassify the complaints of major incidents. In Figure 1, the orange line is the optimal boundary and corresponds to the best F1 value for customer complaints data (Tang et al., 2021). In summary, the Figure 1 demonstrates that the metrics for different applications require models to be biased between minority and majority classes, i.e. this green line as a balanced boundary is not the best for F1 value and G-means. However, existing methods aimed at obtaining balanced models and cannot generalize well to other applications. Although re-weighting methods can tune hyperparameters (i.e. class weights) to control the degree of model imbalance, the large range of values leads to serious time-consuming for searching optimal values. How to make the models efficiently achieve the appropriate imbalance correction becomes a new challenge.

In this paper, we propose a new optimization method that can efficiently adjust the models to specified imbalance states according to application metrics or requirements. This method compensates for the lack of explicitly regulating the model imbalance in existing works, so that the method can be broadly applicable to many scenarios that require a variable degree of model imbalance. Specifically, we first use a class probability distribution to formally define model imbalance state(MIS) which can describe and quantify the imbalance of a model. Then, we propose bias adjustment (BA) method that optimizes the bias of the last layer of a model to make the deformed model reach the optimal MIS in applications. BA is an efficient method because it participates in a simple calculation in the last layer of the model and optimizes a few parameters with the same number of labels. In addition, BA has wide applicability since users can give imbalance metrics to determine the target MIS for BA optimization. Finally, we introduce an overall training strategy which uses the BA method to correct the biased model in every epoch of the traditional training process. An advantage is that the strategy facilitates the discovery of optimal model parameters for representation learning.

In brief, we summarize four main contributions as follows. (1) We give a formal definition of model imbalance state(MIS) so that the imbalance of a model can be precisely quantified. (2) We propose a bias adjustment (BA) method that can efficiently correct the imbalanced models and broadly adapt to different applications based on user-specified metrics. (3) We introduce a training strategy to discover the optimal representation parameters during imbalanced learning. (4) We perform extensive experiments to verify the effectiveness and efficiency of our method.

## 2 BACKGROUND

**Imbalanced Classification.** Let $X = ((x_i, y_i))_{i=1}^N$ be a training set, where $x_i$ is the i-th sample, $y_i$ is the corresponding label and N is the number of samples. In K-category classification, the label $y$ has K possible values which are denoted as $C_1, ..., C_K$. Let $n_1, ... n_K$ be the number of training samples for the class $C_1, ..., C_K$ respectively. A deep learning model $\Phi$ can be viewed as a mapping function from input x to target $\hat{y}$, that is, $\hat{y} = \Phi(x)$, and $\hat{y}$ is also the prediction for input x. The learning goal is to reduce the difference between the prediction $\hat{y}$ and the real label $y$. We use cross-entropy loss function $l(y, \hat{y})$ to measure this difference. Thus, the training objective is to minimize the sum of the losses over the entire training set $X$, formally, the overall loss $L_{CE} = min \sum_{i=1}^N l(y, \Phi(x))$. However, if $\exists\, i, j\; n_i \ll n_j$, then, the training set is imbalanced. The model learned from $L_{CE}$ will be seriously biased toward the majority class, that is, most samples of the minority class $C_i$ may be misclassified into the majority class $C_j$.

**Decoupling Models.** Recently, researchers have proposed to solve the imbalanced classification by decoupling the models. The deep learning models consist of complex computational structures, though it can be simply divided into two parts, that is, the backbone and classifier modules. The backbone module is used to obtain the feature representation of the input, such as a BERT model (Devlin et al., 2018) can extract the representation of the text. We denote $z = f(x; \theta)$ as the representation of the input $x$, where $f$ is the function of the backbone module and $\theta$ is the parameter. The classifier module refers to the last layer of a deep learning model, which takes the representation $z$ as input and outputs the label probability. Generally, the last layer is a linear classifier and we denote $\boldsymbol{W} = \{w_i\}_{i=1}^K$ and $\boldsymbol{b} = \{b_i\}_{i=1}^K$ are the linear weight matrix and bias respectively, where $w_i \in R^d$ and $b_i$ are the weight and bias corresponding to label $C_i$. The probability of $C_i$ is calculated by softmax.

$$\hat{y}_i = \frac{exp(w_i^T z + b_i)}{\sum_{k=1}^K exp(w_k^T z + b_k)} \tag{1}$$

The researchers found that the classifier module was mainly affected by the imbalanced data, rather than the backbone module. Therefore, many works are devoted to adjusting the classifier module to tackle the imbalanced classification(Kang et al., 2019).

**Classifier from Probability Theory.** From the perspective of probability theory, the classifier is the conditional probability $p(y|z)$ that is the probability distribution of labels under the given representation $z$, and the label corresponding to the maximum probability is the predicted result. According to Bayesian formula, the probability of label $C_i$ is

$$p(y = C_i|z) = \frac{p(z|C_i)p(C_i)}{\sum_{i=k}^K p(z|C_k)p(C_k)} = \frac{exp(w_i^T z + b_i' + lnp(C_i))}{\sum_{k=1}^K exp(w_k^T z + b_k' + lnp(C_k))} \tag{2}$$

where assuming $p(z|C_i)$ is the member of exponential family distributions and is further restricted to be a linear function of $z$ in the exponent (Bishop & Nasrabadi, 2006), and $w_i$ and $b_i'$ are the linear parameters of $p(z|C_i)$. Comparing the forms of Eq.(1) and Eq.(2), it can be found that

$$b_i = b_i' + lnp(C_i) \tag{3}$$

It suggests that the estimate of the class probability $p(C_i)$ is included in the bias of the last layer.

## 3 ALGORITHM

In this section, we first give the definition of model imbalance state (MIS) and then propose the bias adjustment method that can efficiently correct the imbalance. Thereafter, we introduce a training strategy for the imbalanced classification.

### 3.1 MODEL IMBALANCE MEASURE

Recently, many works focus on building a class-balanced model, but the balance between minority and majority classes is not always optimal. In fact, the optimal model in different applications has varied degrees of imbalance among classes. To measure the degree of imbalance of a model, we

introduce the concept of model imbalance state (MIS) which is denoted as $P \in R^K$ and $P_i$ records the model prediction probability of label $C_i$. If the value of $P_i$ is large, the model will be biased towards label $C_i$ and become imbalanced. We can estimate $P_i$ from the training set $X$. Formally, given a model $\Phi$ and a dataset $X$, $P_i$ can be obtained by

$$p(C_i|\Phi) = \int p(C_i|x,\Phi)p(x)dx = E(p(C_i|x,\Phi)) \approx \frac{1}{N}\sum_{x \in X} p(C_i|x,\Phi) \qquad (4)$$

Eq.(4) shows $P_i$ can be estimated as the average prediction probability of label $C_i$ on $X$.

## 3.2 BIAS ADJUSTMENT

When the model is trained in a traditional way on the imbalanced dataset, the predicted probability of the majority class in MIS is much greater than that of the minority class. The main idea of this work is to correct the model by adjusting the MIS, such as increasing the minority class prediction probability. Additionally, inspired by Eq.(3), bias contains the estimation of class probability, thus we propose a bias adjustment method (BA) that only adjusts the bias to change the MIS. Specifically, given an expected class probability distribution $r$, BA adjusts the bias $\boldsymbol{b}$ to make the model imbalance state $P$ close to $r$ and uses KL divergence to build this objective $L_{bal}$ as follows

$$L_{bal} = -\sum_{i=1}^{K} r_i ln(\frac{P_i(\boldsymbol{b})}{r_i}) \Leftrightarrow -\sum_{i=1}^{K} r_i ln(P_i(\boldsymbol{b})) \qquad (5)$$

where the right side removes the constant term $\sum_{i=1}^{K} r_i ln(r_i)$ that is independent of $\boldsymbol{b}$.

In practice, the class probability distribution $r$ is generally unknown in applications. BA uses a search strategy to find the optimal $r^*$ which is based on imbalance metrics on a validation set, i.e. F1 value, G-means. The details of this search strategy and bias optimization are as follows.

**Search Strategy.** This work mainly discusses the binary classification, so only the minority class probability $r_1$ needs to be adjusted and the majority class probability $r_2$ can be determined by $r_2 = 1 - r_1$. In order to perform an efficient search, BA finds $r_1^*$ from $(0,1)$ step by step with the precision of 10 powers. BA first finds the best value $a_1 10^{-1}$ where $a_1$ from $\{1,...,9\}$, and then finds the best value $a_1 10^{-1} + a_2 10^{-2}$ where $a_2$ from $\{-9,...,9\}$, and similarly finds $a_1 10^{-1} + a_2 10^{-2} + a_3 10^{-3}$ and $a_3 \in \{-9,...,9\}$. Generally, the precision taken to $10^{-2}$ or $10^{-3}$ is enough.

**Bias Optimization.** BA uses gradient descent to calculate the optimal $b*$ for the objective $L_{bal}$. In fact, this is a simple optimization because $b$ participates in the calculation at the last layer of the model and has only K parameters. Therefore, BA can first store the calculation results in the model feed-forward and adopt the entire training data set as a batch, which can calculate the optimal $b^*$ efficiently and accurately.

## 3.3 A TRAINING STRATEGY FOR IMBALANCED CLASSIFICATION

We introduce a new training strategy that alters the MIS in the traditional training process. This strategy does not need to change the traditional training method and still uses the $L_{CE}$ to update the model parameters. However, after each epoch in the training, the strategy will try to use the BA to correct the bias and validate the model to retain the best model. The motivation is that, the traditional training way on the imbalanced dataset may only slightly affect the backbone module but seriously shift the classifier. Therefore, after training in the traditional way, the strategy keeps the backbone module constant and only adjusts the bias of the classifier. The experimental results indicate that the effect of the bias adjustment is excellent in the binary classification task.

**Discussion.** The training strategy has two advantages for imbalanced classification. First, the strategy is beneficial to discover the best backbone module parameters because it can adjust the imbalance and validate the model at each epoch of training. In contrast, the two-stage methods train a stable model on an imbalanced dataset in the first stage and then balance the classifier in the second stage. It is difficult to ensure that the parameters of backbone module are optimal. Although the two-stage methods can also adjust the classifier at each epoch, the optimization of the classifier is more time-consuming than BA. Second, our strategy enables the model to meet different application requirements. Because the strategy corrects the imbalance based on user-specified metrics, in

Table 1: Statistics of three datasets

| Data sets | Classes | Training Samples | Testing Samples |
|---|---|---|---|
| CIFAR-10 | 2 | $2 \times 5000$ | $2 \times 1000$ |
| SST-2 | 2 | $2 \times 5000$ | $2 \times 5000$ |
| AG | 2 | $2 \times 20000$ | $2 \times 20000$ |

other words, the degree of model imbalance can be determined by the requirement of application. However, traditional imbalanced approaches either obtain class-balanced models that may not be applicable, or suffer from expensive hyperparameter tuning to suit the needs of the application.

## 4 EXPERIMENTS

### 4.1 EXPERIMENTAL SETUP

**Datasets and Evaluation.** We use CIFAR-10 (Schneider et al., 2019) for image classification and adopt SST-2 sentiment analysis data (Socher et al., 2013) and AG news data (Zhang et al., 2015) for text classification. Specifically, we select the class 0 and 1 from CIFAR-10 and the class "World" and "Sci/Tech" from AG to form binary classification datasets. The statistics of the datasets are shown in Table 1. Further, we construct the imbalanced datasets from these three datasets. We set the example ratios of majority class to minority class 10:1, 50:1, 100:1 and 500:1 for the CIFAR-10 and SST-2 datasets, and we set the extremely imbalance ratio of 1000:1 for the AG dataset. In addition, we use 3 metrics to evaluate the results, which are accuracy rate, F1 value of the minority class and G-means. The F1 value comprehensively measures the precision and recall of the minority class, and G-means calculates the geometric mean of the recall of the majority and minority classes (Du et al., 2017). It is noted that the accuracy is evaluated on a balanced test set, and F1 and G-means are calculated on a test set with an imbalance ratio equal to the training set.

**Comparison Methods.** We compare our method with six approaches: (1) **Baseline**, the model is directly trained on an imbalanced training set with cross-entropy loss. (2) **Proportion**, an empirical class weighting method that weights examples by inverse class frequency. (3) **Auto-Weighting**, the method proposed by Hu et al. (Hu et al., 2019) can learn data weights from a small validation set. (4) **cRT**(Kang et al., 2019), a two-stage method that re-trains the classifier with class-balanced sampling at the second stage. (5) **LWS**(Kang et al., 2019), a two-stage method that learns the scaling factors for the classifier at the second stage. (6) **POT**(Guo et al., 2022), is the SOTA approach that considers automatic weighting and the two-stage method. The experimental details are described in Section A.

### 4.2 RESULTS OF DIFFERENT METRICS ON TEXT CLASSIFICATION AND IMAGE CLASSIFICATION

**Results on Accuracy.** The accuracy results on SST-2 and CIFAR-10 are shown in Table 2. There are mainly the following three observations. (1) Our method achieves the best accuracy at different imbalance ratios on datasets SST-2 and CIFAR-10, which shows that just adjusting the bias of the classifier can greatly improve the distorted model due to the imbalanced data. It also implies that the impact of imbalanced learning on the backbone module and the classifier weight parameter may not be severe. (2) When the dataset is more imbalanced, our method is more advantageous than other methods. For example, on SST-2, our method outperforms proportion by only 0.19 accuracy points at 10:1 but 7 points at 500:1, and on CIFAR-10, our method exceeds POT by only about 1 point at 10:1 but more than 3 points at 500:1. The possible reason is that when the number of minority class examples decreases, our method of adjusting the bias is less susceptible to overfitting than optimizing the entire model parameters or the classification module. (3) The performance of the two-stage methods POT and cRT on CIFAR-10 is stable and excellent at different imbalance ratios, but the results on SST-2 are lower than proportion. This may be because the representation learning of the model at first stage is of high quality on CIFAR-10, while it is not optimal on SST-2.

Table 2: Results of accuracy on SST-2 and CIFAR-10 under different imbalance ratios

| Methods | SST-2 | | | | CIFAR-10 | | | |
|---|---|---|---|---|---|---|---|---|
| Imbalance Ratios | 500:1 | 100:1 | 50:1 | 10:1 | 500:1 | 100:1 | 50:1 | 10:1 |
| Baseline | 50.00 | 58.61 | 66.97 | 82.46 | 62.27 | 78.25 | 84.24 | 96.17 |
| Proportion | 57.03 | 79.00 | 83.13 | 87.77 | 68.39 | 81.65 | 87.80 | 96.93 |
| Auto-Weighting | 50.25 | 61.16 | 61.54 | 81.96 | 57.99 | 76.81 | 84.98 | 96.04 |
| LWS | 50.13 | 56.17 | 59.82 | 79.95 | 61.95 | 76.05 | 77.44 | 90.88 |
| cRT | 50.13 | 55.97 | 60.29 | 78.51 | 76.81 | 88.18 | 91.52 | 97.43 |
| POT | 52.41 | 63.78 | 75.40 | 81.95 | 77.10 | 89.11 | 91.72 | 96.42 |
| **Ours.** | **64.45** | **80.31** | **83.76** | **87.96** | **80.89** | **91.72** | **93.98** | **98.04** |

Table 3: Results of F1 value on SST-2 and CIFAR-10 under different imbalance ratios

| Methods | SST-2 | | | | CIFAR-10 | | | |
|---|---|---|---|---|---|---|---|---|
| Imbalance Ratios | 500:1 | 100:1 | 50:1 | 10:1 | 500:1 | 100:1 | 50:1 | 10:1 |
| Baseline | 0.00 | 10.18 | 31.67 | 65.43 | 29.33 | 73.23 | 72.67 | 95.74 |
| Proportion | 4.33 | 17.54 | 34.96 | 65.17 | 34.11 | 72.45 | 75.60 | 96.20 |
| Auto-Weighting | 0.00 | 8.62 | 17.24 | 54.49 | 19.05 | 74.02 | 73.20 | 95.75 |
| LWS | 0.00 | 11.96 | 22.17 | 58.97 | 5.12 | 50.51 | 52.11 | 89.92 |
| cRT | 0.00 | 12.25 | 27.81 | 59.69 | 29.96 | 71.30 | 71.93 | 95.31 |
| POT | 3.84 | 7.72 | 15.05 | 52.80 | 2.40 | 22.76 | 43.66 | 94.71 |
| **Ours.** | **12.15** | **22.42** | **41.36** | **66.79** | **44.64** | **77.66** | **76.89** | **96.24** |

It illustrates that the well-trained backbone module parameters are crucial for two-stage methods, and further analysis is presented in section 4.3.2.

**Results on F1 Value.** The results of F1 value are shown in Table 3. There are two main conclusions. (1) Our method also achieves the best performance on F1 value in all cases. In particular, on SST-2, our method outperforms the second-best by nearly 8 F1 points at 500:1 and 6 F1 points at 50:1, and on CIFAR-10 with the ratio of 500:1, our method surpasses the second-best method by more than 10 points. It demonstrates that our method has the dominant performance on the F1-value metric, and it also suggests that modifying the model imbalance state (MIS) to fit the metric is effective. (2) The two-stage methods POT, cRT and LWS, and the auto-weighting method all hardly work on the F1-value metric. The F1 values of these methods are almost lower than baseline at different imbalance ratios on SST-2 and CIFAR-10. It indicates that learning with the objective on a balanced dataset or class-balanced sampling is not suitable for the F1-value metric. In other words, the MIS suitable for F1 value is more likely to have a small minority class probability, rather than a completely balanced minority class and majority class.

**Results on G-means.** The results of G-means are shown in Table 4, and we can obtain the following two observations. (1) Similar to the results of the F1 value, our method has the best performance on G-means in all cases. Especially on SST-2 dataset, our method exceeds the second-best method by 13 G-means points at 500:1 and 10 points at 100:1. It again illustrates the superiority of our method and the importance of adjusting the MIS to fit the metrics. (2) The performance of the two-stage methods POT and cRT is better than the baseline on the G-means metric. Especially when

Table 4: Results of G-means on SST-2 and CIFAR-10 under different imbalance ratios

| Methods | SST-2 | | | | CIFAR-10 | | | |
|---|---|---|---|---|---|---|---|---|
| Imbalance Ratios | 500:1 | 100:1 | 50:1 | 10:1 | 500:1 | 100:1 | 50:1 | 10:1 |
| Baseline | 0.00 | 22.07 | 53.63 | 80.66 | 34.13 | 80.93 | 80.94 | 97.02 |
| Proportion | 0.00 | 74.00 | 83.82 | 87.62 | 62.31 | 83.14 | 87.26 | 97.94 |
| Auto-Weighting | 0.00 | 48.45 | 59.85 | 83.06 | 28.26 | 80.59 | 81.03 | 97.37 |
| LWS | 0.00 | 33.10 | 52.27 | 77.66 | 72.93 | 74.92 | 68.64 | 90.98 |
| cRT | 0.00 | 41.22 | 52.69 | 78.03 | 93.43 | 94.55 | 94.77 | 98.45 |
| POT | 59.80 | 64.55 | 71.30 | 81.12 | 90.65 | 94.71 | 94.53 | 97.32 |
| **Ours.** | **73.68** | **84.87** | **85.41** | **88.04** | **93.51** | **97.69** | **97.93** | **98.86** |

Table 5: Results of different metrics on AG with the imbalance ratio of 1000:1

| Methods | Accuracy | F1 Value | G-means |
|---|---|---|---|
| Baseline | 52.02 | 3.2 | 6.32 |
| Proportion | 82.78 | 27.22 | 77.25 |
| Auto-Weighting | 82.24 | 24.73 | 76.48 |
| LWS | 70.65 | 17.02 | 49.35 |
| cRT | 83.67 | 4.23 | 79.22 |
| POT | 73.50 | 8.46 | 67.16 |
| **Ours.** | **86.71** | **29.11** | **86.02** |

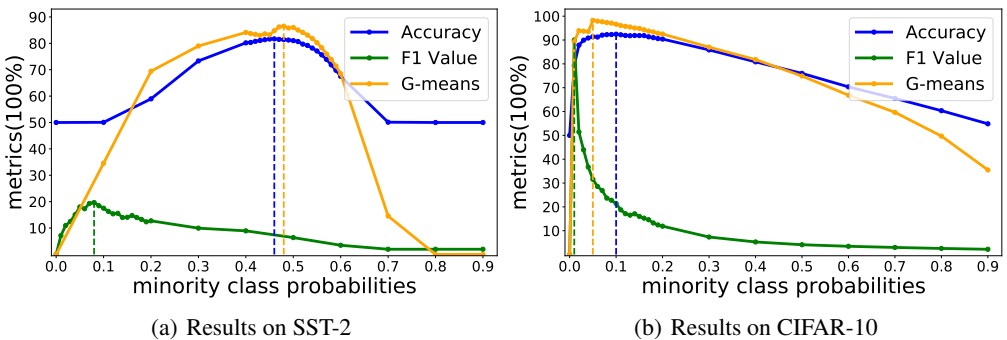

(a) Results on SST-2                       (b) Results on CIFAR-10

Figure 2: Results of three metrics on different minority class probabilities

the imbalance is serious, the baseline method is almost invalid on G-means. It shows that the MIS suitable for G-means requires a greater probability of the minority class than the F1-value metric.

**Results on AG Dataset.** Table 5 shows the results of accuracy, F1 value and G-means on the AG dataset with the extremely imbalanced ratio. We summarize the following two conclusions. First, we can see that our method achieves the best results on all metric, which further proves that the method of modifying MIS to adapt metrics is successful. Second, the experimental results show that our method is also effective in extremely imbalanced conditions.

## 4.3 INSIDE ANALYSIS AND EFFICIENCY COMPARISON

### 4.3.1 INSIDE ANALYSIS

**Optimal MIS on Different Metrics.** To validate that different metrics correspond to different optimal MIS, we present the results of regulating the model imbalance to different MIS. Here, we express the MIS in terms of minority class probabilities. Figure 2 show the results of accuracy, F1 value and G-means on SST-2 and CIFAR-10. We can observe that the probabilities corresponding to the highest values are different among these metrics. Specifically, the probability of the highest F1 value is less than 0.1 on SST-2 and close to 0 on CIFAR-10, which is much smaller than that of G-means and accuracy. This is because the F1 value takes into account the precision and recall of the minority class. If the minority class probability is great, a large number of majority classes will be misclassified into the minority class, which will greatly reduce the precision of the minority class and result in a decrease in F1 value. Thus, a high F1 value may favor a small minority class probability. On the contrary, the highest G-means requires a great minority class probability. Because G-means considers the recall of both the minority class and majority class, and the great minority class probability significantly improves the recall of the minority class, thereby increasing the G-means. For the accuracy, the best probabilities is close to 0.5 on SST-2 and is 0.1 on CIFAR-10, which shows the optimal MIS may also vary on different datasets. In summary, the different imbalance metrics prefer different MIS, and the balanced optimization strategy may be not best, e.g. on the F1 value. It indicates that the work of regulating the model imbalance is necessary and the BA method is effective.

**Impact of Epochs on the Results.** We show the BA correction results at each epoch during the traditional training process. The results of different metrics on SST-2 and CIFAR-10 are shown

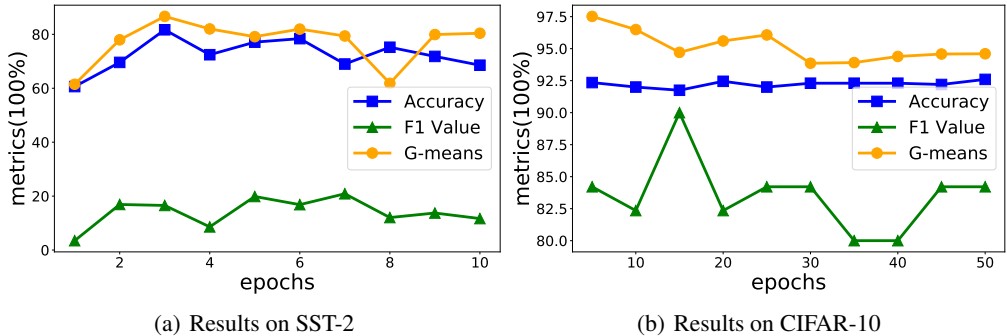

(a) Results on SST-2  (b) Results on CIFAR-10

Figure 3: Results of three metrics at each epoch during training

Table 6: Comparison to the two-stage methods with tuning the epochs on SST-2

| Methods | Accuracy | F1 Value | G-means | time(h) |
|---|---|---|---|---|
| LWS(10th epoch) | 50.13 | 0.00 | 0.00 | 0.25 |
| LWS(best epoch) | 53.12 | 6.04 | 45.36 | 2.34 |
| cRT(10th epoch) | 50.13 | 0.00 | 0.00 | 0.25 |
| cRT(best epoch) | 58.80 | 3.57 | 66.66 | 2.37 |
| POT(10th epoch) | 52.41 | 3.84 | 59.80 | 0.46 |
| POT(best epoch) | 62.54 | 10.30 | 66.74 | 4.44 |
| **Ours.** | **64.45** | **12.15** | **73.68** | **0.30** |

in Figure 3. As the training epoch increases, the results on these metrics are roughly stable but still vary. Especially, this variation is significant on the F1 value of CIFAR-10. It indicates that the number of training epochs may greatly affect the models and the quality of the representation parameters determines the performance. However, in actual implementation, due to the influence of imbalanced data, we cannot explicitly know the optimal representation parameters of the model on which epoch during training. Therefore, we encourage correcting and validating the model on each epoch to obtain the best representation parameters.

### 4.3.2 EFFICIENCY COMPARISON

**Compared to the Two-stage Methods.** We tune the best epoch for the two-stage methods and compare the performance and time consumption with our method. The results are shown in Table 6. We summarize the following three points. (1) The results of the two-stage methods at the best epoch are significantly improved compared to the 10th epoch, which illustrates the importance of the selection of the optimal model parameters. (2) The results of the two-stage methods at the best epoch are still lower than our method, which indicates the effectiveness of our method. (3) The time consumption of tuning the epochs for the two-stage methods is about 10 times that of our method, which shows the efficiency of our method.

**Compared to the class-level weighting Method.** We set the minority class weight to 1 and We use grid search to find the optimal weight value from $(0, 10)$ for the majority class. We sequentially increase the number of weight values according to the exponential of 2, so we tested a total of $2^{11}$-1 and $2^{10}$-1 values for SST2-2 and CIFAR-10, and our method is tested once. This comparison is shown in Figure 4 where the triangles represent the results of our method. We summarize the three points. (1) The time cost of tuning the weights to get good results differs from our method by 2-3 orders of magnitude. For example, in Figure 4(a), the weighting method takes nearly 100 hours and the accuracy is close to 80%, while our method takes less than 1 hour to achieve an accuracy of more than 80%. It demonstrates the clear advantage of our method in terms of efficiency compared to the tuning the weights. (2) The weighting method by tuning a large number of weights is still almost lower than our method on all metrics, which further illustrates the effectiveness of our method. (3) In Figure 4(a), our method is slightly lower than the weighting method on F1 value, which shows that optimizing the entire model parameters can perform better than adjusting the bias.

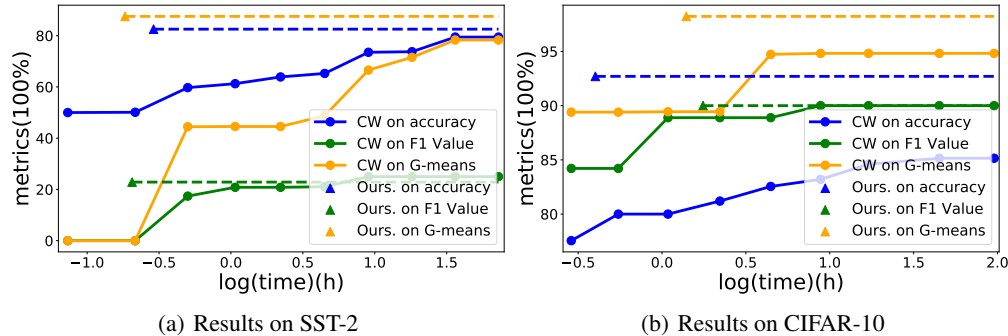

(a) Results on SST-2             (b) Results on CIFAR-10

Figure 4: Comparison to the weighting method with tuning the weight values. These results are on the imbalance ratio of 100:1, and CW and Ours. represent class weighting and our method.

## 5   RELATED WORK

**Re-sampling.** The re-sampling methods obtain the balanced deep models by re-balancing the training data distribution. These methods mainly include up-sampling method of increasing the minority class samples (Chawla et al., 2002; Shi et al., 2022) and down-sampling method of reducing the majority class samples (Drummond et al., 2003; Barandela et al., 2004; Liang et al., 2022) to achieve the balance of the number of samples between classes.

**Weighting Methods.** There are very rich works on weighting samples for imbalance classification, and we summarize into two main groups, namely empirical weighting and automatic weighting. The empirical weighting methods set the manual weight values to the samples, such as inverse class frequency (Wang et al., 2017), inverse square root of class frequency (Mikolov et al., 2013; Mahajan et al., 2018), calculating weight values based on the effective number of examples (Cui et al., 2019), hard example mining (Dong et al., 2017; Shrivastava et al., 2016) and Focal loss (Lin et al., 2017). The automatic weighting methods obtain adaptive weights through learning mechanisms. Ren et al. (Ren et al., 2018) and Hu et al. (Hu et al., 2019) proposed to learn the example weights by a meta-learning paradigm. Similar methods also include the work of Liu et al.(Liu et al., 2021), Meta-weight-net (Shu et al., 2019) and Meta-class-weight (Jamal et al., 2020). Recently, Guo et al. (Guo et al., 2022) proposed a automatic weighting method based on optimal transport (OT). However, these automatic weighting methods are all based on a balanced validation set to learn the sample weights.

**Two-stage Methods.** The two-stage methods focus on representation learning at first stage and rebalance the classifier at second stage, such as OLTR (Liu et al., 2019), LDAM (Cao et al., 2019) and cRT(Kang et al., 2019). Experiments prove the effectiveness of these training strategies for addressing the imbalance problem (Li et al., 2021; Zhong et al., 2021).

The above methods aim to obtain balanced deep models by re-balancing data, learning weights on a balanced data set or balanced optimization strategies. There is a key difference between our work and theirs. We believe that a balanced model may be not beneficial for practical applications, thus, we propose to regulate the degree of model imbalance to promote applicability.

## 6   CONCLUSION

To solve the imbalance problem of different applications, we propose a new optimization strategy that can efficiently regulate the imbalanced deep models based on the user-specified metrics. Thus, this strategy can be widely applied to different scenarios. Specifically, we define model imbalance state and propose the BA method that can efficiently correct the distorted model, and finally we introduce the overall training framework. The experimental evaluation shows that our algorithm can achieve significant improvement compared with the SOTA method in terms of efficiency and effectiveness.

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

## A  APPENDIX

**Experimental Details.** For text classification, we use BERT as the classification model. We adopt Adam optimization and set the learning rate to $4e^{-5}$ and $5e^{-6}$ on SST-2 and AG, respectively, and batch size is 64. For all methods, the BERT is trained for 10 epochs. cRT and LWS re-train the classifier for 50 epochs at second stage, and the learning rate is selected from $\{1e^{-1}, 1e^{-2}, 1e^{-3}\}$. POT uses official code and settings, and epochs are also set to 50 at second stage. The automatic weighting also adopts official code and the decay of data weights is tuned from $\{1, 5, 10\}$. We construct a balanced validation set of 10 samples for POT and automatic weighting. In our method, the learning rate of BA is set to 1.

For image classification, we use ResNet-32 (He et al., 2016) as the classification model. Following Li et al. (2021), we use 200 epochs and set the learning rate as 0.1, which decays to $1e^{-3}$ and $1e^{-5}$ at 160th and 180th epochs. The settings of the two-stage methods including cRT, LWS and POT is the same as for text classification. The model is further trained 50 epochs for proportion and our method and 20 epochs for the automatic weighting. The learning rate of BA is set to 10.

Finally, All experiments were implemented with Python 3.8 and PyTorch 1.8 and were run on a Linux server with RTX 3090 GPU and 128GB RAM. All results are averaged over 5 runs.

