# OpenReview forum: "Regulating Imbalanced Deep Models with User-Specified Metrics"
_ICLR.cc/2024/Conference — Submitted to ICLR 2024_

### Official Review · Reviewer_b3Zi · 2023-10-12

**Soundness:** 2 fair
**Presentation:** 2 fair
**Contribution:** 2 fair
**Rating:** 3
**Confidence:** 4

**Summary:**

This paper introduces a precise definition to quantify the extent of model imbalance and presents a bias adjustment technique that effectively fine-tunes the model for specific imbalance scenarios. The proposed approach involves training the backbone model using conventional training methods and then updating the model's bias term to achieve class balance. The authors demonstrate performance enhancements compared to baseline models on datasets like CIFAR-10, SST-2, and AG. Additionally, the paper discusses additional metrics, such as F1 score and G-means, which previous studies did not address.

**Strengths:**

The strengths of this paper include:

1. It provides a solid mathematical rationale for addressing class imbalance, which lends credibility to the proposed method. The approach is well-grounded in mathematical derivations, reinforcing its validity.

2. The proposed algorithm stands out for its simplicity. It leverages traditional training methods and only requires minimal adjustments, specifically focusing on updating the bias term. This simplicity makes it practical and straightforward to implement.

3. The proposed method consistently demonstrates performance improvements across a range of datasets, further highlighting its effectiveness and versatility.

**Weaknesses:**

Despite the improved performance demonstrated by the proposed method in comparison to other baselines, assessing its overall impact presents certain challenges.

1. The dataset employed in the study does not align with typical choices for class imbalance tasks. For instance, the authors utilize a subsampled version of CIFAR-10, which is not commonly used for this purpose. Larger datasets like the full CIFAR-10 or CIFAR-100 are typically preferred to evaluate the method's compatibility with scenarios involving more than two classes.


2. There is a notable absence of comparison with recent works such as RIDE [1], CMO [2], CUDA [3], Balanced-softmax [4], BCL [5], and NCL[6], which are relevant in the context of class imbalance. These omissions are not addressed in the related work section, leaving a gap in the comparison with contemporary methodologies.

[1] Long-tailed Recognition by Routing Diverse Distribution-Aware Experts, ICLR 2021
[2] The Majority Can Help the Minority: Context-rich Minority Oversampling for Long-tailed Classification, CVPR 2022
[3] CUDA: Curriculum of Data Augmentation for Long-Tailed Recognition, ICLR 2023
[4] Balanced Meta-Softmax for Long-Tailed Visual Recognition, NeurIPS 2020
[5] Balanced Contrastive Learning for Long-Tailed Visual Recognition, CVPR 2022
[6] Nested Collaborative Learning for Long-Tailed Visual Recognition, CVPR 2022

**Questions:**

1. It would be beneficial if the authors could provide a comparison of their method's effectiveness against recent works published within the last two years. This would help establish the relevance and competitiveness of their approach in the current research landscape.

2. Additionally, extending the evaluation to datasets like CIFAR-10/100 without subsampling of classes and including large real-world datasets such as ImageNet or iNaturalist2018 would offer a more comprehensive assessment of the method's applicability and generalization capabilities. This broader range of datasets would provide a better understanding of the method's performance in diverse and real-world scenarios.

**Details Of Ethics Concerns:**

I do not have ethics concerns about this paper.

---

### Official Review · Reviewer_TH73 · 2023-10-28

**Soundness:** 2 fair
**Presentation:** 2 fair
**Contribution:** 2 fair
**Rating:** 3
**Confidence:** 3

**Summary:**

The paper addresses the challenge of imbalanced problems in deep learning. The authors introduce two primary contributions:
1. **Model Imbalance State (MIS)**: A definition to quantify the imbalance of a model. This metric essentially measures the model's prediction probability for a given label.
2. **Bias Adjustment (BA)**: An optimization method that adjusts the bias of the model's last layer to make the imbalanced model reach an optimal state. In addition, BA employs a search strategy for the best bias values and uses gradient descent to find the optimal bias.

Experiments show that their proposed method achieves significant improvements and demonstrates superior effectiveness.

**Strengths:**

1. The paper tackles the prevalent issue of imbalanced problems in deep learning, which is important especially when dealing with real-world data.
2. Experiments show that the bias adjustment method seems computationally efficient.
3. Experiments looks thorough and demonstrate the improvement and effectivenss of proposed method.

**Weaknesses:**

The issue addressed in this paper is undeniably significant, and the authors present an intuitive approach backed by promising results. However, the central section (i.e., sec.3) requires further refinement, as its current state suggests an incomplete work. My concerns are as follows:

1. The statement "$P_i$ records the model prediction probability of label $C_i$" can benefit from a detailed explanation, preferably supplemented by an example for clarity.
2. The assertion "Eq.(4) shows Pi can be estimated..." is misleading, given the absence of $P_i$ in Eq.(4).
3. Regarding Eq.(4), the terms $p(C_i | \Phi)$ and $p(x)$ need explicit definitions within this section. Given that the MIS is formally defined, readers should not be expected to reference prior sections for notation clarification.
4. The portion "In practice, the class probability distribution r is generally unknown in applications. BA employs a search strategy to pinpoint the optimal $r^∗$ rooted in imbalance metrics" raises questions. Specifically, are there any theoretical guarantees to ensure a bounded error between the searched $r^*$? Additionally, are there assurances that the BA method remains effective even if the discovered $r^*$ lacks accuracy? Expanding on this could fortify the paper's foundation, preventing it from being solely experimental.
5. Section 3.3 is somehow sparse. I recommend elaborating on the process and possibly incorporating a figure for enhanced comprehension.
6. If the problem pertains to multi-class classification rather than just binary, would the proposed method still be applicable? This aspect warrants discussion in your paper, particularly as your focus is on addressing imbalanced datasets found in real-world scenarios, which are often not limited to binary classification.
7. Is there potential for this method to be extended to address imbalanced regression scenarios? For instance, rather than having the model predict strict values of 0 or 1, could it be adapted to regress responses ranging from 0 to 1? It might be beneficial to address this point in the paper, even if it falls outside the scope of Section 3.

In conclusion, I urge the authors to consider these suggestions earnestly. Improvements to section 3 could greatly elevate the overall quality of this paper.

**Questions:**

See above

---

### Official Review · Reviewer_HH3t · 2023-10-30

**Soundness:** 3 good
**Presentation:** 3 good
**Contribution:** 4 excellent
**Rating:** 6
**Confidence:** 4

**Summary:**

This paper explores the backbone-and-classifier bifurcation of a neural network to propose a change in the way that biases are assimilated into the decision. In doing so, an observation on the class predictions by a learner on an imbalanced classification is made, that the majority class dominates the predictions, which is reported as a measure. This is absorbed into a bias correction mechanism that optimizes  relative entropy between the distribution of predictive bias and the ideal bias with no imbalance. The later quantity being unknown, an approximative step is introduced following every training  epoch.

**Strengths:**

The proposed method is straightforward. Evaluations are however, fairly extensive, and are done across a variety of data types - tabular, image and text. A per-measure comparison is also provided. This kind of papers are persuasive with broad-based evaluations, usually on even more datasets.

**Weaknesses:**

The hypothesis of just the bias on the softmax layer meant for classification making the classification swing by these amounts us astounding. In fact, a study on the values of the bias versus predictive performance is required in my opinion before bias adjustment is presented.

**Questions:**

At 500:1 imbalance ratios, multiple benchmark methods show numbers decimated to zero. A previous study, MMM, by Mirza et al. '21 appears to suggest that even at such a ratio, classical resampling and even baselines report an above zero performance. Could you reason about the disparity?

---

### Official Review · Reviewer_tgc4 · 2023-11-08

**Soundness:** 3 good
**Presentation:** 2 fair
**Contribution:** 3 good
**Rating:** 6
**Confidence:** 4

**Summary:**

This paper presents a solution to an important issue of class imbalance observed frequently during classification. Imbalanced classification problems are generally handled using decoupled models where we have a backbone module that learns the input feature representation and a classifier module that takes the learned feature representation and performs the classification task. In this work, authors have developed user-specified metrics as an optimization strategy to resolve class imbalance in classification tasks. Towards this, they formally define the Model Imbalance State (MIS) based on how biased the classifier is towards the particular class and used Bias Adjustment (BA) method to minimize the bias of the classifier towards the minor class based on MIS value, which in general is biased towards the majority class. Using BA, authors have employed the KL divergence-based objective to minimize the difference between the class probability distribution observed during MIS and the original class probability distribution using a naive search strategy and optimization using gradient descent. Unlike traditional training strategies to handle imbalanced data classification where the backbone module is taken care of, this work keeps the backbone module constant and works on bias adjustment to minimize the impact of class imbalance over classification performance. The proposed approach is evaluated over binary text and image classification datasets compared with six approaches, including class weighting, two-stage methods, and a SOTA approach. The results are reported using the accuracy, minority class F1, and G-means evaluation metrics. The results reported under different imbalance ratios suggest that the proposed approach is capable of handling the imbalance scenarios better than all the compared approaches.

**Strengths:**

1. The idea of BA to minimize the difference between MIS and original class distribution to achieve better classification results over imbalanced datasets sounds very intuitive.
2. The proposed approach is able to handle the extreme imbalance scenarios better.
3. The proposed approach is more efficient than two-stage methods in terms of the time taken to tune the epochs and is ten times faster than SOTA.
4. The proposed approach is much more efficient than class-level weighting methods, where the time taken for tuning the weight values takes as much as 2-3 orders of magnitude than the proposed approach.

**Weaknesses:**

1. The evaluation metrics chosen to report the performance reflect the overall performance rather than being class-specific, except for F1 over the minority class. Towards this, authors should have considered precision, recall, and F1 value over minority class and macro performance utilizing the same metrics to check if the performance is not biased towards any particular metrics or a particular class, respectively.
2. The results on CIFAR-10 as can be observed from Figure 3(b) reflects a high variance in F1 value. What could be the possible reason behind this?

**Questions:**

1. Why have authors considered only BERT and ResNet-32 for text and image datasets, respectively, out of so many models widely available?
2. Can we expect the same behavior over other backbone models? Why?

---

### Meta-Review · Area_Chair_PyrQ · 2023-12-09

**Metareview:**

This paper was reviewed by four experts and received 6, 6, 3, 3 as the ratings. The reviewers concurred that the paper addresses an important problem, the proposed method is simple and efficient, and extensive evaluations have been conducted on a wide variety of data types (tabular, images and text). However, they mentioned that the proposed method has not been compared against recent baselines, which is necessary to establish the relevance and competitiveness of the approach in the current research landscape. It was also mentioned that the proposed method has been evaluated only on binary classification problems; it needs to be validated on multi-class problems, as real-world imbalanced datasets are often multi-class. Further, experiments need to be conducted on larger datasets like CIFAR-100, ImageNet and iNaturalist2018 to comprehensively assess the method's applicability and generalization capabilities.

The authors did not upload a revised version of their paper, and also did not respond to the individual reviewer's comments. In light of the above discussions, we conclude that the paper is not ready for an ICLR  publication in its current form. While the paper clearly has merit, the decision is not to recommend acceptance. The authors are encouraged to consider the reviewers’ comments when revising the paper for submission elsewhere.

**Justification For Why Not Higher Score:**

The reviewers raised concerns about the experimental evaluations, mainly regarding comparison baselines and multi-class datasets. The authors did not upload a revised version of their paper, and also did not respond to the individual reviewer's comments. This does not merit acceptance of the paper at ICLR, considering its high standards.

**Justification For Why Not Lower Score:**

N/A.

---

### Decision · Program_Chairs · 2024-01-16

Reject